# Cell Membrane Features as Potential Breeding Targets to Improve Cold Germination Ability of Seeds

**DOI:** 10.3390/plants11233400

**Published:** 2022-12-06

**Authors:** Lakhvir Kaur Dhaliwal, Rosalyn B. Angeles-Shim

**Affiliations:** Department of Plant and Soil Science, Davis College of Agricultural Sciences and Natural Resources, Texas Tech University, Lubbock, TX 79409-2122, USA

**Keywords:** cold stress, cotton, fatty acid, membrane lipids, unsaturation

## Abstract

Cold stress breeding that focuses on the improvement of chilling tolerance at the germination stage is constrained by the complexities of the trait which involves integrated cellular, biochemical, hormonal and molecular responses. Biological membrane serves as the first line of plant defense under stress. Membranes receive cold stress signals and transduce them into intracellular responses. Low temperature stress, in particular, primarily and effectively affects the structure, composition and properties of cell membranes, which ultimately disturbs cellular homeostasis. Under cold stress, maintenance of membrane integrity through the alteration of membrane lipid composition is of prime importance to cope with the stress. This review describes the critical role of cell membranes in cold stress responses as well as the physiological and biochemical manifestations of cold stress in plants. The potential of cell membrane properties as breeding targets in developing strategies to improve cold germination ability is discussed using cotton (*Gossypium hirsutum* L.) as a model.

## 1. Introduction

Climate change due to global warming is one of the grand challenges of the 21st century. The unprecedented rate by which the planet is heating up continuously destabilizes climate towards supercharging weather extremes in the form of heat waves, drought, large storms and colder and longer winters that heavily impact agriculture. Although global warming tilts the odds in favor of more hot days, plant ecologists predict the higher risks of chilling injury to crop plants as the occurrence of extremely cold weather become more common in agricultural regions worldwide [1]. Indeed, there has been an increasing number of reported cases of the devastation caused by unseasonably cold weather in agricultural production in recent decades. In 2008, for example, the unusually harsh winter that hit southern, central and eastern China destroyed 6.87 million hectares of wheat and vegetable crops [2]. In Western Australia, a record-breaking cold spell in 2018 caused serious frost damage to wheat and barley that were estimated to bring in 14 million tons of harvest that year [3]. In 2021, a winter storm that hit Texas devastated citrus and vegetable crops in the state, resulting in at least US$380 million in losses [4].

Several studies have shown that germinating seeds, particularly of tropical crops such as cotton [5], maize [6], sorghum [7] and rice [8], are highly sensitive to cold stress [9]. Imbibition of cold water induces excessive leakage of solutes, including amino acids and proteins that are conducive to pathogenic spore germination on seeds [10]. In soybean, higher leakage of soluble sugars has been reported to enhance the occurrence of damping off due to Pythium infection [11,12,13]. Pathogenic infections combined with cold-induced seed tissue disintegration ultimately lead to poor, slow and non-uniform germination. This translates to poor crop performance in terms of yield and quality [14]. In maize, delayed germination and poor emergence have been shown to lower plant density, leading to uneven stand establishment and reduced yields [1,15,16,17,18].

The complexity of mechanisms and pathways regulating low temperature stress responses in plants, especially during germination, presents a big challenge in developing cold-tolerant crop cultivars [18]. With technological advances in biology and biochemistry, however, several studies have successfully identified cell membranes as the first line of defense against cold stress [18,19,20]. Manifestations of cold-induced damages in germinating seeds including metabolic dysfunction, cell death and embryo abortion have been linked to perturbations in the structure and function of the cell and organelle membranes. These alterations in cell membrane properties also serve as stress sensors that trigger calcium influx and downstream responses to cold [21]. This review summarizes the protective functions of cellular membranes during the very early stage of germination. Using this compiled information as a backdrop to the thesis of the paper, we highlight important features in the chemical structure of the membranes that can potentially be targeted to improve seed germination under cold stress, using cotton as a model. 

## 2. Role of Cell Membranes in Early Germination 

Seed germination is the first developmental phase in the life cycle of higher plants and is highly determinant of the ecological and economic productivity of crops [22,23]. It is defined by a specific sequence of events that begins when a dry, mature seed takes in water and terminates when the radicle protrudes through the seed coat. During the whole process, the seed continuously takes in water in a characteristic tri-phasic pattern. The first phase is known as imbibition and is defined by a rapid influx of water into the seed through the chalazal aperture. The water then passes through the nucellar tissue around the embryo, then to the radicle tip (Figure 1a) [24]. As the seed hydrates, cellular events that move the seed from a quiescent to a metabolically active state are initiated. The first critical consequence of imbibition is the structural re-configuration of the cell membranes [25,26]. 

Membranes serve as a stable barrier between the internal and external environments of the cell. They are composed primarily of a lipid bilayer that is semi-permeable in nature. The predominant membrane lipid components of the bilayer, otherwise known as glycerophospholipids or membrane lipids, have a hydrophilic phosphate head and two hydrophobic fatty acid chains attached to a glycerol molecule. In dehydrated tissues such as mature seeds, the hydrophilic head groups orient themselves to surround whatever water molecules remain in the seed, while the hydrophobic tails are directed away from the water [9,26,27]. This inverted arrangement of phospholipids in dry, mature seeds is known as the hexagonal II conformation and is characteristically susceptible to cellular leakage compared to the bilayer arrangement (Figure 1b). As soon as imbibition begins, the cell membranes of mature seeds quickly revert to their original, selectively permeable, bilayer configuration to prevent excessive leaking of solutes outside the cell. When seeds are sufficiently hydrated to ≥50% of their dry weight, the activity of enzymes necessary for cell organelle differentiation, breakdown of storage reserves and transport to growing tissues and *de novo* mRNA synthesis and translation are upregulated [28]. As the seed enters a metabolically active state, the net gain in water content becomes negligible. This defines the second phase of water uptake. The last and third phase of water uptake is marked by a secondary increase in hydration as the embryonic root protrudes through the seed coat and aids in water absorption [29] (Figure 1c). 

The speed of membrane reorganization depends upon the flexibility of cell membranes under specific conditions. A highly flexible cell membrane can quickly reorganize from hexagonal II to lamellar configuration, thus preventing excessive solute leakage. However, cold conditions tend to induce close packing of membrane lipids, resulting in membrane rigidification. The ability of membranes to maintain flexibility in response to low temperature is highly regulated by fatty acid unsaturation content and chain length [30,31,32,33]. The unsaturated fatty acids are generated by enzymes known as fatty acid desaturases (FAD) that are present in the chloroplastic and endoplasmic reticulum membranes [33]. *FAD* genes convert saturated fatty acids to unsaturated fatty acids by creating double bonds between the carbon molecules. Mechanisms explaining the biochemistry of double bonds have been described extensively in other studies [34,35]. The highly reactive double bonds, an important feature of unsaturated fatty acids, prevent the close packing of hydrophobic chains, thereby lowering phase transition temperature and keeping the membrane flexible. In plants, a total of seven *FAD* genes that differ in their localization and function have been identified [33,36]. Although all the *FAD* genes are membrane-bound, the first double bond is created by a soluble *FAD* known as *stearoyl-ACP desaturase* (*SAD*) which converts stearic acid (18:0) to oleic acid (18:1) in an acyl-carrier protein (ACP)-bound form. Among the membrane-bound *FAD* genes, *FAD2* and *FAD6* convert oleic acid (18:1) to linoleic acid (18:2) and are in the ER and plastids, respectively [37]. The creation of a third double bond is a function of *FAD3*, *FAD7* and *FAD8* which synthesize linolenic acid (18:3) in the endoplasmic reticulum (*FAD3*) and chloroplasts (*FAD7*, *FAD8*). The production of palmitoleic acid (16:1) is mediated by the plastid-bound *FAD* genes, namely *FAD4* and *FAD5*, which specifically target palmitic acid (16:0) bound to glycerol (PG) and diacylglycerol head groups, respectively. *FAD2-1*, a sub-class of *FAD2*-, has been identified as a seed-specific desaturase that synthesizes linoleic acid (18:1) in the young seeds and developing buds of plants [38,39]. Cold-induced *FAD* gene expression has been widely studied in a number of crops such as maize [40], rice [41] and soybean [42]. In rice, *FAD*-mediated increases in oil unsaturation under cold stress significantly decreased oxidative damages caused by reactive oxygen species during germination [40].

Aside from fatty acid unsaturation and chain length, the type of polar head groups present in the membrane lipids also affect membrane properties, and thus its germination ability, under cold stress. Depending on the type of polar head group present, glycerophospholipids can be of six types viz., phosphatidic acid (PA), phosphatidylcholine (PC), phosphatidylethanolamine (PE), phosphatidylglycerol (PG), phosphatidylinositol (PI) and phosphatidylserine (PS). PA and PE have smaller head groups relative to the width of their fatty acid chains. The overall ratio of the head to the tail group in these two glycerophospholipids tends to give them a conical shape (Figure 2). Conversely, all other glycerophospholipids, namely PC, PS, PG and PI, possess a head group that is similar in width to their fatty acid chains. This gives these sets of membrane lipids a generally cylindrical shape. The abundance of cylindrical lipids provides membranes with a stable lamellar configuration, whereas the conical lipids tend to encourage a negative curvature of membranes towards a leaky hexagonal II configuration. Under normal conditions, the relative proportions of different glycerophospholipids are balanced. However, under cold stress, enzymes such as phospholipases are activated as a signaling response to the stress. Phospholipases hydrolyze cylindrical glycerophospholipids such as PC, PG and PI to produce PA. The overproduction of PA tends to give membranes a leaky hexagonal II configuration. In soybean, phospholipase-mediated PA production has been reported as the major cause of imbibitional chilling injury. Inhibition of the activity of PLD significantly improved the germination performance of soybean under cold stress [9]. 

Cell membranes play key roles in abiotic stress signaling. Under cold conditions, enzymes such as phospholipases, kinases and phosphatases are activated to mediate the production of the lipid-signaling molecule, PA. There are two distinct pathways to phospholipase-mediated synthesis of PA in seeds. The first involves the hydrolysis of the membrane lipids PC, PE, PG and PS by phospholipase D (PLD) to generate PA [21,44,45]. The second involves the hydrolysis of PI by phospholipase C (PLC) to generate diacylglycerol (DAG) and inositol triphosphate (IP3). DAGs resulting from PLC-mediated hydrolysis of PI is phosphorylated by diacylglycerol kinase 2 to produce PA. Additionally, the non-specific PLC (NPC), an isoform of PLC, targets membrane lipids such as PC and PE, but not PI, to produce PA. PA and IP3 produced from membrane hydrolysis play direct roles in ion homeostasis and hormone signaling, particularly abscisic acid (ABA), in response to cold stress [21,46]. In *Arabidopsis,* for example, insertional mutants of the *lysophosphatidic acid acyltransferase* gene that catalyzes the conversion of PA to DAG resulted in PA accumulation and strong hypersensitivity to ABA during germination. These results provide evidence of the direct role of PA in ABA signaling [47]. 

Cold-induced increases in the local concentration of PA create ideal docking sites for enzymes where several proteins get recruited from the cytoplasm to the cell membrane for membrane-bound functions [48]. In rice, the direct binding of MAP kinase 6 and sumoylated E3 ligase to the PLDα-generated PA in the membrane has been shown to enhance chilling tolerance [49]. The PLC/DGK pathway has also been reported to be activated under cold stress conditions [50].

Given that cell membrane homeostasis is a prerequisite for the normal germination and physiological functioning of the seed, cell membrane dynamics present a viable target for the improvement of cold germination ability in crops. 

## 3. Cellular Responses to Cold-Induced Stress in Cell Membranes

Under low temperature stress, membrane lipids transition from a liquid crystalline phase to a gel phase [25]. Membrane lipids in a gel phase pack tightly together and therefore have lower lateral mobility. In contrast, membranes in a crystalline phase are fluid due to the more random orientation and loose packing of the glycerophospholipids. The cold-induced close packing of membrane lipids during germination has direct implications in the role of cell membranes in maintaining cellular integrity, molecular transport, cellular signaling and respiration [1]. 

*Morpho-physiological and biochemical manifestations*. Leaching of solutes outside of the cell is the first measurable effect of cold stress on cell membranes during germination [51]. Perturbations during membrane re-organization cause cellular leakage under normal conditions. Under cold stress, solute leakage is exacerbated by the reduced flexibility of the cell membrane that slows down its conformational transition from the hexagonal II to a lamellar arrangement. The excessive loss of cellular substances including carbohydrates, amino acids, unsaturated fatty acids and various metabolic compounds compromises the various biological processes occurring within a seed, resulting in germination failure [52]. Additionally, the seed leachates provide a rich substrate for bacterial and fungal pathogens that can cause rotting of seeds [53,54]. 

Membrane rigidification also causes lesions that need to be repaired through the synthesis and incorporation of new material into the plasma membranes [55,56]. Prolonged exposure to cold, however, inhibits the synthesis and/or incorporation of new material into membranes, further exacerbating cellular leakage [56]. In *Arabidopsis* and bell pepper, a significant reduction in membrane lipid fractions of the cell was associated with severe damage in membrane structure under cold stress [22,57]. A separate study on *Thellungiella salsuginea* demonstrated the maintenance of membrane integrity and stability as a key factor behind the higher tolerance of the species to low-temperature stress [58]. 

Aside from cellular integrity, the cold-induced reduction in membrane flexibility has also been shown to affect physiological and biochemical processes within the cell. Recent studies in *Escherichia coli* established a close correlation between membrane fluidity and the normal functioning of the electron transport chain in the mitochondria for cellular respiration [59]. Ubiquinone is an important mobile component of the electron transport chain that carries electrons among the enzyme complexes in the mitochondria. Under cold stress, these enzyme complexes are able to maintain their function but the diffusivity of ubiquinone is hampered by the reduced fluidity of the mitochondrial membranes [60]. The decreased diffusion of ubiquinone due to cold stress negatively impacts respiration rates and several energy-driven germination events. 

Lastly, poor development of chloroplast membranes under cold stress have been shown to impede functions related to carbohydrate generation and lipid biosynthesis in the elaborate and continuous network of membranes known as thylakoids [51,61]. 

*Lipid peroxidation through the function of reactive oxygen species.* Seed germination, like many other developmental processes in plants, requires oxygen, which acts as the final receptor of electrons in the electron transport chain (ETC) reactions in the mitochondria. The reduction and oxidation reactions in the ETC of mitochondrial membranes create electrochemical potential, which is required for ATP generation. The final product of these reactions is reduced oxygen, which reacts with hydrogen ions to generate water molecules. Limitations in the supply of oxygen during water imbibition cause leakage of electrons from the ETC. The leaked electrons attack oxygen molecules, resulting in the generation of reactive oxygen species (ROS) in the form of superoxide (O_2_^−^), hydrogen peroxide (H_2_O_2_), hydroxyl radical (^.^HO) and singlet oxygen (^1^O_2_) [62,63]. Under normal conditions, low levels of ROS produced during imbibition promote germination. For instance, H_2_O_2_ lowers ABA content and its transport from cotyledons to embryo, resulting in the mobilization of reserves necessary for germination [64]. In *Arabidopsis* and tomato, ROS facilitates the loosening of cell walls that aid in the emergence of the radicle [65]. Exogenous application of H_2_O_2_ has been reported to improve germination in a number of crop species [64].

With cold stress, rigidification of membranes, including that of the mitochondrion, impedes the diffusivity of the mobile electron carrier, ubiquinone, resulting in the low levels of this molecule in the ETC [60]. Consequently, it slows down the transport of electrons, leading to ROS overproduction and cell death in cold exposed seeds. In its defense, the cell scavenges overproduced ROS and free radicals by activating antioxidant enzymes such as superoxide dismutase, ascorbate peroxidase, catalase, glutathione peroxidase, monodehydroascorbate reductase, dehydroascorbate reductase, glutathione reductase and glutathione S-transferase. Transgenic tobacco overexpressing a *glutathione peroxidase* gene exhibited better metabolic activities relative to the wild types under cold stress [66]. When ROS production goes beyond the scavenging capacity of the antioxidant system, ROS starts attacking cellular constituents such as nucleic acids, proteins and lipids. Free radicals such as hydroxyl ions attract hydrogen from polyunsaturated fatty acids of lipids, resulting in the production of lipid peroxyl radicals and hydroperoxides along with a water molecule [67]. This process, known as lipid peroxidation, destroys the structure of the cell membrane, resulting in metabolic dysfunction and accumulation of toxic compounds in seeds. Malondialdehyde (MDA), the final product of lipid peroxidation, has been widely used as an indirect measure of cold sensitivity [51,68].

*Negative regulation of germination through hormonal signaling.* Germination is regulated by the antagonistic relationship between the plant growth regulators, gibberellic acid (GA) and abscisic acid (ABA) [69]. The balance of these two phytohormones is controlled by external stimuli such as cold and light. During imbibition, environmental cues that are species-specific act in concert to promote germination by decreasing ABA and increasing GA content [70,71]. Given that GA is a positive regulator and ABA is a negative regulator of germination, GA:ABA ratio has been reported to increase three- and ten-fold, respectively, during the early and late phases of germination [28]. This is in accordance with the role of GA in inducing the production of cell wall-remodeling enzymes during both phases of germination [69]. Aside from GA and ABA, other major phytohormones are known to regulate seed germination, including brassinosteroids, ethylene, jasmonic acid and salicylic acid [72,73]. 

Cold stress elicits hormonal signaling that is not limited to the action of a single plant growth regulator but involves different phytohormones acting antagonistically or synergistically with each other [74]. Studies have demonstrated that cold stress triggers ABA synthesis [75] and transport [76] in seeds. Under normal conditions, ATP-binding cassette (ABC) transporters facilitate the transport of ABA from the site of its synthesis (i.e., endosperm) to the site of its action (i.e., embryo). These transporters are highly dependent on ATP and the characteristics of the plasma membrane [76]. 

Cold-induced loss in membrane fluidity upon water imbibition inhibits the activity of ABC transporters. To control ABA homeostasis under cold stress in *Arabidopsis* seeds, HSP70-16 protein, a heat-shock protein induced by salicylic acid under cold conditions, interacts with voltage-dependent anion channels (VDAC) to keep them open [76,77,78]. Open VDAC channels facilitate the efflux of ABA from the endosperm to the embryo, resulting in the delay in germination [79]. The transported ABA in the embryo triggers the hydrolysis of membrane lipids such as phosphoinositol, which amplifies cold signaling in seeds [79]. Similarly, cold stress is also known to trigger the synthesis of ethylene and jasmonic acid [80,81]. In wheat, jasmonic acid triggers cold germination by suppressing ABA production [82]. In *Arabidopsis*, ethylene improves cold germination by inhibiting ABA signaling [41]. An auxin signaling repressor known as IAA8 acts as a positive regulator of seed germination in *Arabidopsis* as it downregulates the ABI3 regulator involved in ABA signaling [83]. Both cold and ROS are responsible for the accumulation of auxins in seeds [83]. 

An overview of the seed responses to cold stress at the germination stage is presented in Figure 3. 

## 4. Modifying Cell Membrane Properties towards Enhancing Cold Germination Ability: A Case Study in Cotton

Because cell membrane integrity and cold stress tolerance are closely associated, changes in the cell membrane properties must be critical for improving cold tolerance at the germination stage. The insertion of ‘single-action’ genes involved in complex cellular pathways tends to stabilize cell membranes towards the improvement of abiotic stress tolerance in plants. Transgenic potato plants expressing the *Lipid Transfer Protein 1* gene showed enhanced membrane integrity, which reduced electrolyte leakage and improved chlorophyll content under multiple abiotic stresses [84]. The potential of improving cell membrane attributes to enhance cold germination ability is discussed in the succeeding paragraphs, using cotton as a model. 

Cotton is a global commodity that is grown primarily as a major source of natural fiber, as well as oil and feedstock. It is cultivated in over 60 countries worldwide, occupying approximately 29.3 million hectares of land [18,85]. Despite the successful introduction of cotton to a wide range of eco-environments, its tropical/subtropical origin makes it innately susceptible to cold stress at all stages of its life cycle. Previous studies have shown that cotton germination is optimum at a temperature range of 28–30 °C, whereas juvenile to adult vegetative growth is best at a temperature range of 21–30 °C. Temperatures below the cardinal minimum of 15 °C (which constitutes a physiological equivalent of 0 °C for cotton) impede the overall growth and development of the plant [86,87]. 

The inherent sensitivity of cotton to cold stress severely restricts the planting window for the crop, especially in temperate regions where growing seasons are short. To avoid cold spells that can cause irreversible damage to the plant at the early growth stages, planting dates are often delayed until the average weekly temperature no longer falls below 15 °C. The practice of delaying planting until the weather is warmer ensures an optimum temperature for seed germination, although it effectively extends the cultivation window beyond the range of temperature that is not optimum for cotton at later stages of growth (e.g., boll maturation) [88,89]. Exposure of mature cotton plants to the cooler temperatures and shorter days of fall can negatively impact fiber yield and quality due to cold-induced reductions in cell wall thickening and delay in fiber elongation [90]. As an alternative, the planting season for cotton can be shifted to an earlier date. Early-season planting guarantees that the crop will mature under warmer temperatures, although it risks subjecting the seeds to colds snaps that are lower than the cardinal minimum requirement for germination. In spite of the challenges associated with early-season planting, farmers remain interested in the practice as a means to exploit residual moisture in the soil from winter precipitation and to take advantage of the reported benefits of this cultural management on cotton yield [91]. Several studies have shown that early-season planting increases the number of flowers per plant, boll weight and leaf area index of cotton plants, leading to a 10–15% increase in yield compared to plants grown at normal dates [92,93]. 

Considering the potential yield gains from early-season planting, various seed treatments such as hydropriming and application of osmo-protectants have been used to alleviate the negative effects of low-temperature stress on germination. While these strategies successfully provided seeds with a degree of protection against low-temperature stress during germination, the additional expenditures associated with the treatments take away from the overall profitability of planting cotton early in the season. In the long term, the most economical and efficient strategy to establish the production stability of cotton planted early will be to develop and cultivate varieties with enhanced tolerance to cold stress during germination. More importantly, determination of the underlying mechanisms that confer adaptive cold stress responses to the cotton seed during germination will be necessary in developing a more precise breeding platform that specifically targets the improvement of traits associated with cold germination ability. The following sections provide an overview of cell membrane properties that have been targeted for improvement as a means to enhance the cold germination ability of cotton seeds.

*Engineering fatty acid composition in cotton seeds.* Cottonseed oil is highly unsaturated and is composed of four major fatty acids (FA), including linoleic acid (18:2) (52.89%), palmitic acid (16:0) (25.39%), oleic acid (18:1) (16.35%) and stearic acid (18:0) (2.33%) [94,95]. FA synthesis in cotton seeds starts one day post-anthesis and ends 60 days post-anthesis. Between 1 and 30 days post-anthesis is when FA synthesis is most variable [96]. 

The chloroplasts and the endoplasmic reticulum (ER) are the two major sites of FA biosynthesis in cotton seeds. In the chloroplast, FAs are produced starting from a two-carbon compound known as acetyl CoA and elongate into a 16-carbon chain bound to an acyl carrier protein (C16:0-ACP) with the addition of 2 carbons at a time. C16:0-ACP is converted to C18:0-ACP by the activity of the *KASII* enzyme. The C18:0 molecule undergoes unsaturation by the addition of a first double bond by *stearoyl acid desaturase* (*SAD*) to create C18:1. The three fatty acids, palmitic acid, stearic acid and oleic acid, then move to the ER for the formation of polyunsaturated fatty acids such as linoleic acid and linolenic acid via *FAD2* and *FAD3*, respectively. 

FA composition has been a major target of various transgenesis and mutagenesis research directed towards increasing the unsaturation content of cotton oil [97,98]. Chemically induced cotton mutants with low palmitic but higher linoleic and linolenic acid content have been shown to be robust germinators even at a critically low temperature stress of 12 °C [5]. Physiological assays have shown that the same mutants with higher degrees of unsaturation were able to maintain normal rates of water uptake and had lower electrolyte leakage compared to cultivars with lower unsaturation content when germinated at 12 °C (data not shown). In contrast, cotton mutants with higher palmitic acid content but lower linoleic acid content performed poorly, similar to the conventional cultivars imbibed at the same temperature. 

Similarly, a transgenic cotton line overexpressing the *FAD2-4* gene and having higher linoleic acid exhibited greater seedling vigor under cold stress compared to the wild-type. In addition, the overexpression line exhibited better overall growth performance in terms of leaf area, fresh weight and plant height under a low temperature stress of 20/15 °C [99,100]. 

The presence of even one double bond can decrease the melting temperature of fatty acids from 69.3 °C (18:0) to 13.4 °C (18:1), ultimately decreasing the cumulative melting point of seeds [25] and thus phase transition temperatures. By altering the fatty acid composition of cellular membranes, a number of studies have confirmed the major role of unsaturation content in maintaining membrane flexibility. The cell membranes with higher unsaturation content facilitate quicker membrane reorganization during the first phase of water uptake, thereby avoiding excessive solute leakage, which leads to better germination performance [5]. Besides the role of fatty acid composition in membrane fluidity, it also contributes to cold tolerance by altering the function of membrane-bound proteins [101,102]. H+/ATPase activity has been reported to be enhanced by an increase in unsaturation content under cold stress [103]. H+/ATPase is a multi-functional integral protein which creates electrochemical gradients and provides energy for other membrane transporters across the cell membrane. Similarly, unsaturation content of chloroplast phospholipids such as PG contributes to the proper functioning of D1 protein under cold stress [104]. This protein plays a key role in repairing the stress-induced photoinhibition damage to photosystem of thylakoid membranes. Chloroplast biogenesis that starts during the second phase of germination has been reported to be highly regulated by the polyunsaturation content in thylakoid membranes under cold stress [105].

Modifying the unsaturation content is thus an efficient strategy to maintain membrane integrity and cellular functioning under cold stress tolerance at the germination stage.

*Engineering membrane lipid content and composition in cotton seeds.* Membrane lipids may be in the form of PA, PC, PE, PG, PI and PS. During germination under normal conditions, imbibing seeds start synthesizing membrane lipids necessary for membrane rearrangement, replacement and repair [27,106,107,108,109]. This process is also important for membrane biogenesis and assembly to support the development of mitochondria and plastids, which are scarce in dry seeds but are important for energy production and metabolism during germination [110]. 

Cold water imbibition in a sensitive cotton cultivar have been determined to impair the ability of the cell to repair membrane damage caused by the stress (data not shown). This, combined with the decreased fluidity of cell membranes and extensive production of ROS, can intensify membrane destruction, resulting in excessive solute leakage and eventually cell death [26,27,111,112]. The total phospholipid content in a cell has been used as a direct measure of cold tolerance [113]. Genotypes that can synthesize membrane lipids during imbibition have been reported to have better tolerance to cold stress during germination.

Aside from lipid content, composition is also a determining factor in cold stress responses. Cold imbibition activates phospholipase enzymes, which hydrolyze membrane lipids such as PC, PE, PG, PI and PS to produce PA [21,44,45,114]. Phospholipase-mediated PA production has dual functions in signaling and maintenance of membrane structure. As a signaling molecule, it interacts with effector proteins including kinases and phosphatases and mediates important physiological processes such as polarized cell growth, osmotic balance, anti-cell death, ABA activation and chilling sensitivity. In a previous study on cotton, PLD-mediated PA has been reported to play direct roles in fiber elongation and development [96]. Under cold stress, increases in PA fractions at the expense of the structural lipids PC, PG and PI exacerbates solute leakage, leading to poor germination under cold stress. This might be due to the tendency of different phospholipids to organize membranes differently. An increase in PA that is accompanied by a decrease in PC, PG and PI tends to configure membranes in a hexagonal II arrangement. Given the leaky nature of this membrane configuration, seeds tend to leak solutes in excess, resulting in poor germination. In cotton, inhibition of phospholipase-mediated PA production under cold stress reduced cellular leakage and improved germination performance of seeds (data not shown). 

Lastly, the ratio of PC to PE has also been used as an important regulator of cold tolerance, with a higher PC:PE ratio providing better cold germination ability [115]. Adjusting the absolute and relative amounts of membrane lipids is thus an efficient approach to avoid the cold-induced membrane injuries.

## 5. Conclusions and Perspectives

Cell membranes have both protective and signaling functions in response to cold stress. The diversity of cell membrane structure and function is highly determined by the fatty acid content and composition of lipids that make up the membranes. For instance, unsaturated fatty acids act as membrane modulators, energy sources, signaling molecules, enzyme activators and resistant barriers under low-temperature stress. This makes lipid content and composition a potential target to improve cold germination ability of seeds. 

Molecular mechanisms underlying the responses of crops to a range of abiotic stresses have considerable overlap in terms of signaling pathways. These overlaps provide the basis for cross-tolerance in plants against stresses like cold, drought and heat. This phenomenon of cross-tolerance indicates the possibility that increasing fatty acid unsaturation will not only confer cold germination ability in seeds, but also potentially improve its tolerance to other environmental challenges. In oilseed crops like cotton, manipulating lipid content towards increased unsaturation has the additional benefit of improving oil quality derived from cotton seeds. This, in turn, will contribute to a more sustainable production of cotton, especially given the agro-climatic challenges in crop production. Additionally, engineering membrane properties of sub-tropical crops such as cotton has the potential to extend its cultivation in areas with a longer winter season. 

## Figures and Tables

**Figure 1 plants-11-03400-f001:**
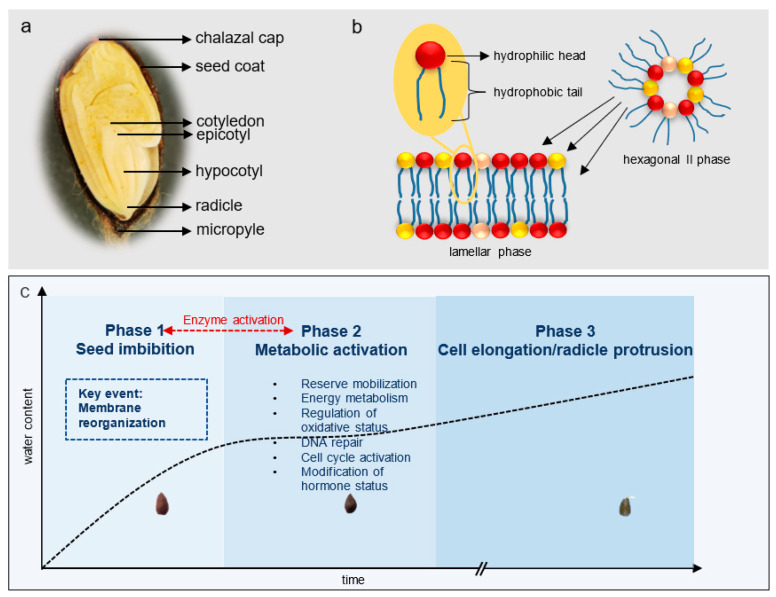
Cross-section of the cotton landrace Hopi seed showing the point of water entry during imbibition (**a**). Illustration of membrane lipids in the lamellar and hexagonal II configuration (**b**). Triphasic pattern of water uptake. The patterns of water uptake at phases I, II and III are indicated by the black dashed line. Key cellular and metabolic processes occurring during each phase of water uptake are described. Membrane reorganization from the hexagonal II to lamellar configuration occurs at phase 1 of water uptake (**c**).

**Figure 2 plants-11-03400-f002:**
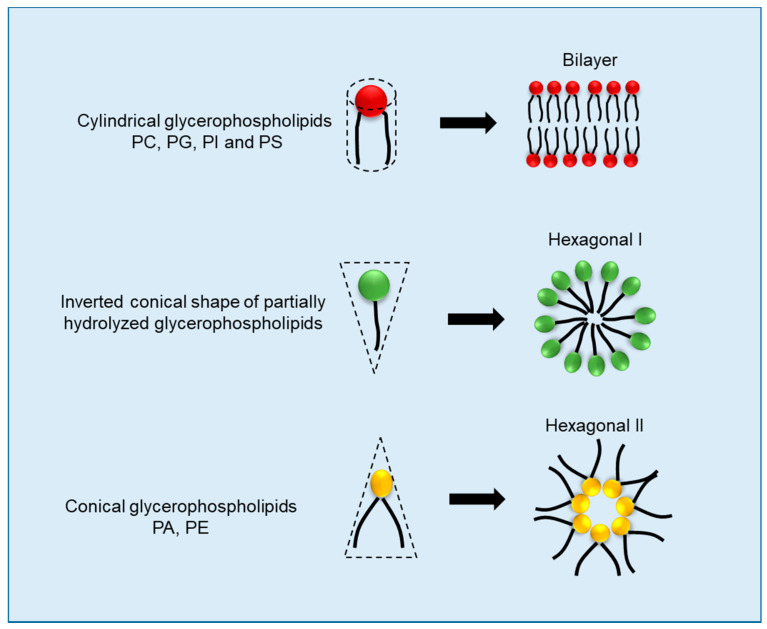
**Cell membrane configurations as affected by the lipid composition.** The relative width of the polar head group and fatty acid chains determines the shape of glycerophospholipids. Cylindrical lipids configure membranes in a bilayer shape, inverted conical in hexagonal I, and conical glycerophospholipids in hexagonal II. Diagram is adopted from Zhukovsky et al. [43].

**Figure 3 plants-11-03400-f003:**
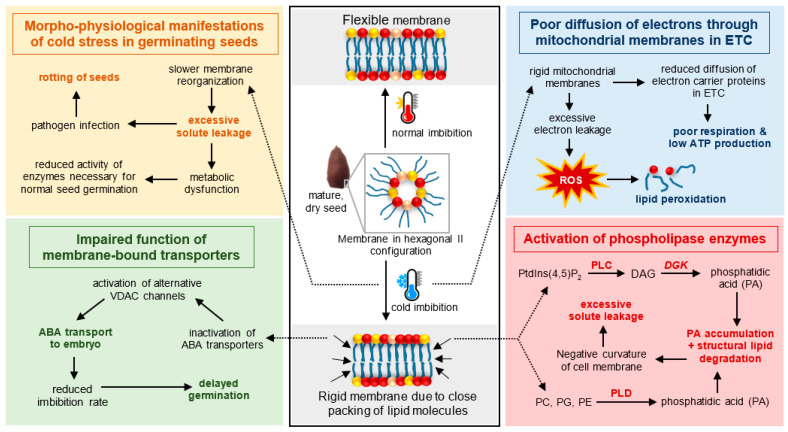
**An overview of the seed responses to cold stress at the germination stage.** Center panel illustrates the process of membrane reorganization from hexagonal II to lamellar configuration during water imbibition under cold and normal conditions. Cold-induced loss in membrane fluidity impairs the process of membrane reorganization and generated downstream responses, which has been explained at the physiological, biochemical and molecular level in the figure.

## Data Availability

This manuscript does not include any data.

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
