# Peer review of "Cell Membrane Features as Potential Breeding Targets to Improve Cold Germination Ability of Seeds"

_plants, 2022, doi:10.3390/plants11233400_

Round 1

Reviewer 1 Report

The title of the work indicates that the authors' intention was to focus on cotton, but they devote the first part of the manuscript to general facts related to the influence of coldness on the cell membrane in plants. The authors should decide whether they devote the article to the general information about the cell membrane during germination in cold conditions - changing the title of work and the second part of the work about cotton or focus on the problem related to cotton - changing the first part of the manuscript. They recommend the major revision article.

Reviewer 2 Report

In this review from Dhaliwal and Angeles-Shim “Targeting cell membranes to improve cold germination ability in upland cotton (Gossypium hirsutum L.)” the authors center their attention on cell membrane protective roles during early germination stages in cold stress comparing to different plant species, enhancing the specific qualities and strategies adopted in cotton at cell physiology, biochemical, hormonal and signaling level to cope with this specific stress. I found overall the review well written and complete, in my opinion of interest for the plant scientific community, my only concern is how the review is concluded and organized, there are no future perspective or personal consideration along the manuscript. This is very evident in the latest paragraph: Engineering membrane lipid content and composition in cotton seeds were a good description taken from the available literature of the current state of art is present, but then suddenly stops, without discussing the concepts or highlighting the current open questions. I suggest the authors to apply these improvements to the manuscript .

Reviewer 3 Report

The manuscript presents interesting and valuable review of recent understanding molecular mechanisms, related to biological membranes, that underlies response to cold in plants (with the focus on cotton seed germination stage). The authors perform not only a thorough analysis of biochemical and biophysical membrane features in context of the plant response to the chilling stress but also the involvement of membranes in signalling, hormonal regulation. Perspective of genetical engineering manipulation to improve plant tolerance to chilling is also outlined.

The manuscript is well written, easy to read and follow since basic knowledge is also provided.

Excellent illustrations help understanding the text.

I can see that this version shows large improvements that makes it almost acceptable for publication.

My concern applies solely to the term "rigid" massively used throughout the manuscript.

We should consider biological membranes as extremely flexible, fluid-like structures which may undergo alteration in fluidity, or compliance or rigidity (as inverse of fluidity) 

Membrane never becomes rigid (in in the common sense) because it always remains fluid-like, otherwise it would not showed semi-permeability or enabled vesicular transport.

see for example  https://en.wikipedia.org/wiki/Membrane_fluidity

So I propose use terms "become more rigid', rigidify, undergo rigidification, increase in rigidity etc.  instead of "rigid" in most cases.

A better term is "Fluidity" (increase/decrease of fluidity) is a better term

Example of correct terminology:

line 338-340 ....affect respiration. Recent studies on Escherichia coli established a close correlation between membrane fluidity and the normal functioning of the electron transport chain in mitochondria for cellular respiration (Gohrbandt et al 2022)"

For example 

line 2016

instead ...  resulting in rigid membranes. I propose ...resulting in membranes more rigid

495 ...Center panel illustrates cold-induced rigidity in cell

..... should be rather .. rigidification...

Cold impairs cellular functions by "too large rigidifying/rigidifaction"

line 631 instead "avoid rigidity"  .... rather "avoid rigidification/ too large rigidification"

Please survey whole manuscript in the respect of often occurring  misuse of term "rigid" .

There are many more similar cases.

Other point

Rigidity of lipids ? .. I my opinion we can refer the term rigidity to a supramolecular structures (resulting from  non-covalent bonding between compounds, e.g. molecules) but not to a single molecule of lipids where there are only covalent bonds between atoms)

line  311 phase (Sanchez et al 2019). Membrane lipids in gel phase are rigid due to the lower lateral mobility 

(gel is less or more rigid rather due to (or better "associated to" ) lower or higher mobility of molecules.)

line 329

"The rigidity of membrane lipids also causes lesions that needs to be repaired through the.."  

Round 2

Reviewer 1 Report

Thanks to the authors for the answer. I understand that cotton is a less studied crop in terms of cell membrane responses to cold stress at germination stage, therefore I believe the authors should write two articles: one about the cell membrane during germination in cold conditions, and the other as a mini-review focused on the problem related to cotton. I recommend the article in its current form be rejected.
